# Additive Effects of L-Arginine with Potassium Carbonate on the Quality Profile Promotion of Phosphate-Free Frankfurters

**DOI:** 10.3390/foods11223581

**Published:** 2022-11-10

**Authors:** Chuanai Cao, Yining Xu, Meiyue Liu, Baohua Kong, Fengxue Zhang, Hongwei Zhang, Qian Liu, Jinhai Zhao

**Affiliations:** 1College of Food Science, Northeast Agricultural University, Harbin 150030, China; 2Heilongjiang Green Food Science & Research Institute, Harbin 150028, China; 3Institute of Advanced Technology, Heilongjiang Academy of Science, Harbin 150001, China

**Keywords:** L-Arginine, potassium carbonate, additive effects, phosphate-free frankfurters, quality profile, sensory evaluation

## Abstract

The present study investigated the additive effects of L-Arginine (L-Arg) with potassium carbonate (PC) on the quality characteristics of phosphate-free frankfurters. The results showed that L-Arg combined with PC could act as a viable phosphate replacer by decreasing cooking loss and improving the textural properties of phosphate-free frankfurters (*p* < 0.05), mainly because of its pH-raising ability. Moreover, L-Arg could assist PC in effectively retarding lipid oxidation in phosphate-free frankfurters during storage (*p* < 0.05). Furthermore, 0.1% L-Arg combined with 0.15% PC was found to exhibit the best optimal phosphate-replacing effect. This combination could also overcome quality defects and promote the sensory attributes of phosphate-free frankfurters to the maximum extent. Therefore, our results suggest that L-Arg combined with PC can be considered a feasible alternative for the processing of phosphate-free frankfurters with an improved quality profile and superior health benefits.

## 1. Introduction

Food-grade phosphates of various chemical forms (e.g., orthophosphates, pyro-phosphates, and polyphosphates, etc.) have been broadly used as synthetic food additives in the meat industry to manufacture meat products, mainly due to their positive effects on the quality profile, sensory attributes, and shelf life of the final products [1]. Phosphates have a unique characteristic of synergistically functioning with sodium chloride to increase the solubility of salt-soluble myofibrillar proteins, subsequently promoting the cooking yield and gel properties of meat products [2,3]. Moreover, phosphates can efficiently raise pH values to alkaline levels and increase ionic strength, eventually increasing the water-holding capacity of meat products [4]. Phosphates also possess the ability to split the complex of actomyosin formed during rigor mortis mainly through the sequestration of calcium or magnesium cations [5]. Furthermore, because of their excellent antioxidant potential, phosphates can effectively retard lipid oxidation, while protecting the flavor and color of meat products during storage [6]. However, with the increasing demand and consumption of processed meat products in many countries, the daily intake of phosphates has become more than two- to threefold higher than intake in the 1990s [7]. The excessive intake amounts of phosphates from processed meat products contributes to an increased risk to health. For example, as reported by Leon, Sullivan, and Sehgal [8], the daily intake of phosphates has already exceeded 1200 mg in some countries, which was significantly higher than the recommended daily intake (700 to 800 mg) by the Food and Agriculture Organization (FAO). Karp, Ekholm, Kemi, Hirvonen, and Lamberg-Allardt announced that a high dietary phosphorus intake could increase the risk of cardiovascular disease, chronic kidney disease, and a poorer bone status [9]. Zhang et al. [10] also indicated that increased phosphorus consumption could significantly break the balance of calcium to phosphorus ratio, resulting in hypercalcemia or hypocalcemia. Consequently, due to their nutritional drawbacks and negative connotation, phosphates have been seriously considered as components that must be partially or completely replaced from the formulation of meat products [11]. However, simply reducing or eliminating phosphates from meat products may compromise the quality profile of the final products. In the last decade, some available phosphate replacers (e.g., non-meat proteins, native or modified starches, hydrocolloids, fibres, seaweeds, plant or vegetable powders, carbonate salts, and alkaline electrolyzed water) and novel processing technologies (e.g., power ultrasound processing and high-pressure processing) have attracted considerable interest for the processing of phosphate-free or reduced-phosphate meat products, with focus on specific quality profiles, including water-holding capacity, emulsion stability, textural properties, colour, sensory attributes, and oxidative stability [7]. However, the commercial viability of these approaches for creating processed meat products with reduced or removed phosphates has been very limited because of the higher manufacturing costs of the final products, particularly in the case of some relatively low-price emulsified meat products (e.g., frankfurters, Vienna sausages, bologna, hot dogs, and meatloaf). Thus, the way to promote the quality profiles of reduced phosphates or phosphate-free processed meat products without increasing the manufacturing cost is challenging for the meat industry.

Carbonates (e.g., sodium or potassium carbonate (PC)), which are classified as acidity regulators, have been used for the processing of meat products, mainly due to their alkalization effects [12]. Xiong indicated that the alkaline carbonates effectively increased the net negative charge of myofibrillar proteins and then promoted the swelling degree of myofibrils, subsequently increasing the water-holding capacity of myofibrillar proteins during thermal treatment [13]. Moreover, because their size is smaller than that of phosphates, carbonates can more easily penetrate the muscle tissue and interact with higher amounts of side chains of meat proteins, significantly promoting the repulsive forces among meat proteins and resulting in higher water retention [14]. In the last decade, numerous concentrated studies have reported that carbonates could be used as phosphate replacers to prepare marinade solutions for enhancing the quality profile (e.g., color, tenderness, water-holding capacity, and flavor) of fresh meat or aquatic products [15,16,17]. Currently, the increasing market trend or consumer demand for manufacturing phosphate-free meat products (e.g., sausage, meatball, meatloaf, and ham) has accelerated the use of phosphate-replacing carbonates by numerous researchers and food industry stakeholders. Furthermore, considering sodium reduction in meat products, PC has been thought to be a preferred phosphate replacer, more so than sodium carbonate. However, our previous, unpublished work indicated that higher concentration (more than 0.20%, *w*/*w*) of PC rendered some negative effects on the quality profile of phosphate-free frankfurters, such as increased porous sections (induced by carbon dioxide during heating treatment) and an unacceptable darker color, which are considered undesirable visual attributes for consumers. Kaewthong and Wattanachant also suggested that a higher concentration of carbonates led to some adverse effects on the function and integrity of actomyosin [14]. However, a lower concentration of PC did not exhibit the optimal substitution effect. Thus, more research is required to identify some suitable techno-functional additives to combine with PC to completely replace phosphates from meat products.

Most recently, some basic amino acids, such as L-Arginine (L-Arg) and L-Lysine (L-Lys), have attracted interest in the meat industry, mainly attributed to their ability to enhance the water-holding capacity, textural properties, and physical/oxidative stability of meat products [18]. In the study of Li et al., they reported that L-Arg or L-Lys promoted the molecular unfolding of myosin and obtained more negative charges, ultimately enhancing the solubility of myosin [19]. Zhang et al. also announced that L-Arg promoted the dissociation of actomyosin and improved the tenderness of chicken breast more efficiently than L-Lys [20]. Thus, we hypothesized that L-Arg possesses the ability to combine with PC and act as a potential phosphate replacer for improving the quality profile of phosphate-free frankfurters. However, to date, there is no available information on this aspect. Therefore, the present study aimed to evaluate the additive effects of L-Arg with PC on the quality profile, flavor attributes, and sensory properties of the resulting phosphate-free frankfurters.

## 2. Materials and Methods

### 2.1. Materials

Post-rigor lean pork shoulder meat (moisture 73.90%, protein 21.36%, and fat 3.17%, based on total weight) and pork back fat (moisture 7.46% and fat 86.95%, based on total weight) were purchased from Gaojin Meat Co., Ltd. (Harbin, China), and were stored on ice while being transported to the laboratory and used on the same day. Food-grade PC (99.5%) and L-Arg (99.5%) were kindly provided by Wangbang Industrial Co., Ltd. (Zhengzhou, China). Composite phosphates (mainly containing sodium tripolyphosphate, sodium hexametaphosphate, and sodium pyrophosphate) were provided by Hens Food Co., Ltd. (Xuzhou, China). The pH value of 0.4% composite phosphate solution (prepared with pH 7.0 ultrapure water) was 9.21 ± 0.02. All spices were kindly provided by McCormick Ltd. (Shanghai, China). Collagen casing (20 mm in diameter) was provided by Qian-ao Co., Ltd. (Harbin, China). All other chemicals and reagents used in the experiments were of analytical grade.

### 2.2. Preparation of Frankfurters

The formulations of frankfurters in each group are shown in Table 1. The phosphate content in the control group was 0.4% (*w/w*, based on the total weight of lean pork meat, pork back fat, and ice). The phosphate-free group (PF group) with no added phosphate was considered the negative control group. Moreover, L-Arg and PC were added to the meat batter of the PF group as follows: T0 group (0.15% PC), T1 group (0.15% PC + 0.025% L-Arg), T2 group (0.15% PC + 0.05% L-Arg), T3 group (0.15% PC + 0.075% L-Arg), T4 group (0.15% PC + 0.1% L-Arg), and T5 group (0.15% PC + 0.125% L-Arg). According to the experimental design, three independent batches of samples (replicates) were prepared on different days, and different treatments in accordance with different formulas were performed for each batch. For each treatment per replicate batch, the meat batter was stuffed into collagen casings (diameter 20 mm) to produce frankfurters with approximate total weights of 2.00 kg (about 70–80 g per sausage). The whole processing procedure of frankfurters was according to the same method described by Choi et al. [21]. Finally, all the frankfurters were vacuum-packed and stored at 4 °C in a refrigerator until further instrumental analyses and sensory evaluation.

### 2.3. Proximate Composition Analysis

The moisture, protein, fat, and ash content of each frankfurter was measured according to the AOAC guidelines [22].

### 2.4. Measurement of PH

The pH values of frankfurters from each group were determined according to the method of Choe et al. [4] with some modifications. Briefly, 10.0 g of each frankfurter was mixed with 90.0 mL of ultrapure water (pH 7.0) and then homogenized by using an IKA high-shear homogenizer (T25 basic, Staufen, Germany) at 5000 rpm for 15 s. Following this, each homogenate was filtered through a Whatman No. 4 filter paper. The pH of the filtrate was then measured by using a pH metre (PB-10, Sartorius Scientific Instruments Co., Ltd., Beijing, China) at room temperature (20–22 °C).

### 2.5. Cooking Loss

The cooking loss (%) was determined by using the same method of Jiang, Cao, Xia, Liu, and Kong [23].

### 2.6. Emulsion Stability

The emulsion stability was determined according to the same procedure as described by Cao et al. [24].

### 2.7. Color

The color of the samples was determined by using the method as described by Chen et al. [25] with a ZE-6000 colourimeter (Nippon Denshoku, Kogyo Co., Tokyo, Japan). The color parameters were expressed as the *L**-value (lightness), *a**-value (redness/greenness) and *b**-value (yellowness/blueness), respectively. This colourimeter had a D65 light source, a 10° observer with an 8-mm-diameter measuring area, and a 50-mm-diameter illumination area. The total color difference (*ΔE**) was calculated by using the following equation:$${\Delta E}^{*}=\sqrt{{{(L}_{0}{}^{*}{-L}^{*})}^{2}{+(a}_{0}{}^{*}{-a}^{*}{)}^{2}{+(b}_{0}{}^{*}{-b}^{*}{)}^{2}}\;$$
where, the values of *L*_0_*-, *a*_0_*-, and *b*_0_*-values were obtained from the control samples, and the values of *L**-, *a**-, and *b**-values were measured from the phosphate-free frankfurters prepared with varying amounts of L-Arg and PC.

### 2.8. Texture Analysis

Before analysis, all the samples were maintained at room temperature (20–22 °C) for at least two hours. Two deformation tests (TDT) were used to evaluate the texture of the frankfurters by using a TA-TX plusC texture analyzer (Stable Micro Systems Co., Ltd., Godalming, UK), which equipped with a P/2 cylindrical probe (diameter 2.0 mm) under the “Hold & Penetration” mode according to the same testing parameters and testing procedure as described by Cao et al. [26].

### 2.9. Electronic Nose Analysis

Electronic nose determinations were performed by using the same method of Zhang, Hu, Wang, Kong, and Chen [27] with a PEN 3.5 electronic nose (Win Muster Airsense Analytics Inc., Schwerin, Germany). The specific types of volatile organic compounds (VOCs) that could be detected by each sensor are shown in Table 2.

### 2.10. Electronic Tongue (E-Tongue) Analysis

E-tongue measurements were performed according to the procedure of Yin et al. [28] with the SA402B E-tongue (Insent Company, Atsugi-Shi, Japan) with five test sensors and two reference sensors.

### 2.11. Assessment of Lipid Oxidation

Lipid oxidation was expressed as the content of malonaldehyde (MDA) and determined in accordance with the method of Wang and Xiong [29] with slight modification. Briefly, 2.0 g of each frankfurter was mixed with 3.0 mL of 1.0% thiobarbituric acid (TBA) solution and 17.0 mL of 2.5% trichloroacetic acid-hydrochloric acid (TCA-HCl) buffer and then vortexed by using a vortex mixer (3030A, Scientific Industries, INC., Bohemia, NY, USA). After that, each of the mixture was heated in the boiling water bath for 30 min and cooled to the room temperature (22–25 °C). Then, 4.0 mL of the above supernatant was mixed with 4.0 mL of chloroform and centrifuged at 3000 rpm for 10 min. The absorbance of supernatant was measured at 532 nm. The thiobarbituric acid reactive substances (TBARS) value was calculated as follows:$$\mathrm{TBARS}/(\mathrm{mg}/\mathrm{kg})=\frac{A_{532}}{m}\times 9.48\;,$$
where, A_532_ represented the absorbance at 532 nm, m represented the weight of the sample (g), 9.48 was the constant.

### 2.12. Sensory Analysis

Sensory evaluation was performed by using the same procedure and testing parameters as those used by Yuan et al. [30]. The sensory evaluation was carried out under the approval permission of the Ethics in Research Committee of Northeast Agricultural University (ERCNEAU-2018-001). All the panelists agreed voluntarily to participate in the sensory evaluation.

Before sensory evaluation, all the frankfurters were maintained at room temperature (25 °C) for two hours. Briefly, 16 panelists (comprising eight females and eight males) with a wealth of experiences in meat products sensorial evaluations were chosen from a pool of graduate students. The detailed selection method of the panelists and preparatory session for the panelists were according to the same methods of Zhang et al. [31]. The sensory tests were performed through three independent sessions with the same panelists. Frankfurters from each group were evaluated by each panelist for each session through a sensory descriptive analysis based on the following parameters on a 7-point descriptive scale: for interior color of frankfurters (1 = yellowness; 7 = pinkness), for degree of uniformity among frankfurters (1 = low; 7 = high), for juiciness of frankfurters (1 = extremely dry; 7 = extremely juicy), for flavor intensity of frankfurters (1 = less intense; 7 = most intense). The final vocabulary and keywords for the interpretation of the attributes can be found in Table 3.

### 2.13. Statistical Analyses

Three independent batches of frankfurters (replicates) were prepared on different days. For each batch, measurements of related traits were performed in triplicate. All data are expressed as the mean ± standard deviations (SD). To evaluate the significance of the main effects (*p* < 0.05), one-way analysis of variance (ANOVA) along with Duncan’s multiple comparison was performed via SPSS Statistics 25.0 software (SPSS Inc., Chicago, IL, USA). ORIGIN software (2018, 64 bit) was used to assess the results of electronic nose and e-tongue measurements based on principal component analysis (PCA).

## 3. Results and Discussion

### 3.1. Proximate Composition

The proximate compositions (e.g., moisture, protein, fat, and ash contents) of frankfurters are shown in Table 4. The PF group exhibited significantly lower protein, water, fat, and ash contents compared to the control group (*p* < 0.05). Desmond reported that phosphates could increase the ionic strength of the meat system and promote the solubilization of myofibrillar protein by the synergistic effect with sodium chloride, which contributed to improving the water-holding capacity in meat products [32]. Huang et al. also indicated that phosphates increased the ionic strength and pH, which contributed to the dissociation of actomyosin into actin and myosin, thus retaining more water in myofibrillar protein [33]. Moreover, the incorporation of 0.15% PC alone (T0 group) remarkably increased the protein, water, and fat contents of phosphate-free frankfurters (PF group) (*p* < 0.05), even obviously lower than that of the control group, which only contained phosphates (*p* < 0.05). LeMaster et al. reported that PC could increase the pH and ionic strength, which exposed charged areas to water which binds to the myofibrillar proteins, thus separating the myofibrils by repulsive forces to enhance the water-holding capacity [17]. Meanwhile, when L-Arg was added to in combination with PC, the protein, water, and fat contents of PF group gradually increased with the increasing addition level of L-Arg compared with that of the T0 group (*p* < 0.05). However, no significant differences for ash contents were noted among T0, T1, T2, T3, T4, and T5 group (*p* > 0.05). Similar results were observed in the study of Silva et al., who reported that the addition of L-Arg increased the protein and fat contents of low-sodium bologna-type sausages [34]. This phenomenon was in accordance with the result of cooking loss and emulsion stability.

### 3.2. PH Values

The changes in pH values of frankfurters from various groups are shown in Figure 1. As expected, the removal of phosphates significantly decreased the pH value of the PF frankfurters when compared with the control group (*p* < 0.05), which was most likely the consequence of the described effect of phosphates on the pH values of meat products. Moreover, the incorporation of 0.15% PC alone (T0 group) remarkably enhanced the pH values of phosphate-free frankfurters (PF group) (*p* < 0.05), showing no differences compared with the pH values of the control group, which only contained phosphates (*p* > 0.05). Our results indicated that PC effectively enhanced the pH values of frankfurters at lower levels in contrast to phosphates. LeMaster et al. reported that PC is the conjugate base of a weak acid (pKa = 10.25) with a strong base and buffering capacity, which efficiently increased the pH values of meat products [17]. Furthermore, when L-Arg was added to combine with PC, with the addition levels of L-Arg increased from 0.025% to 0.125%, the pH values of phosphate-free frankfurters gradually increased from 6.62 to 6.83 (*p* < 0.05). Even on adding the lowest level of L-Arg (T1 group), the pH value was clearly higher than that of the control group (*p* < 0.05). Zhou, Li, Tan, and Sun also reported that L-Arg could deviate the pH values of pork meat from its isoelectric point, subsequently significantly increasing the pH values of pork sausage [35]. Thus, our results indicated the additive effects of L-Arg with PC generated a stronger alkaline power to completely replace phosphates and then satisfied the effect of phosphates on pH values of frankfurters. Additionally, the pH-enhancing ability of the combination of PC and L-Arg can justify the quality improvement of phosphate-free frankfurters.

### 3.3. Cooking Loss and Emulsion Stability

Cooking loss is a crucial indicator that is associated with the water- or fat-binding capacities of frankfurters during heating and subsequent cooling treatments. As shown in Table 5, the PF group exhibited a higher cooking loss than the control group (*p* < 0.05), revealing that more water or fat was released from the meat protein matrix upon the removal of phosphates. Choe et al. indicated that emulsion-type sausages without phosphates had a higher cooking loss (21.31%) than those with 0.3% phosphates [4]. Resconi et al. also suggested that adding phosphates to emulsified meat products promoted the extraction and solubility of myofibril proteins to form a stronger gel network, thereby increasing the water-holding capacity of the resulting emulsified meat products and reducing their cooking loss [36]. Hence, phosphate removal rendered more serious quality defects in frankfurters. Moreover, the incorporation of 0.15% PC alone (T0 group) remarkably decreased the cooking loss of phosphate-free frankfurters (PF group) (*p* < 0.05), even obviously lower than that of the control group which only contained phosphates (*p* < 0.05). LeMaster et al. [17] indicated that the application of 0.3% PC significantly decreased the cooking loss of pork loin chops compared with the application of 0.3% phosphates. Meanwhile, when L-Arg was added in combination with PC, the addition of higher levels (0.075%, 0.1% and 0.125%) of L-Arg significantly lowered the cooking loss of the PF group compared with that of the T0 group (*p* < 0.05). Some previous studies have indicated that L-Arg effectively decreased the cooking loss of meat products [35,37]. Thus, the decreased cooking loss of phosphate-free frankfurters in our present work was closely associated with the enhancement of pH values induced by the additive effects of L-Arg with PC.

As shown in Table 5, the total released liquid, including released fat and released water of the PF group were remarkably higher than that of the control group (*p* < 0.05), which implied that the absence of phosphates obviously decreased the emulsion stability of the frankfurters. Moreover, the incorporation of 0.15% PC alone (T0 group) obviously decreased the total released liquid of phosphate-free frankfurters (PF group) (*p* < 0.05), showing no differences compared with that of the control group (*p* > 0.05). Meanwhile, when L-Arg was added to in combination with PC, the addition of higher levels (0.1% and 0.125%) of L-Arg resulted in significantly higher emulsion stability of phosphate-free frankfurters in comparison with that of the T0 group (*p* < 0.05), as indicated by the obviously lower values of total released liquid, including released water and released fat. As mentioned above, higher pH values induced by the combination of L-Arg and PC effectively increased the net negative charges of meat proteins and enhanced their water-holding capacity, subsequently decreasing the exudation of water or fat [38]. Roldán, Antequera, Pérez-Palacios, and Ruiz also indicated that the increased pH value of meat promoted larger gaps between myofilaments through electrostatic repulsion and then reinforced interactions between myosin and water, which subsequently contributed to increasing the emulsion stability of the meat batter [39]. Furthermore, the guanidinium group of L-Arg positively participated in inhibiting proteins aggregation [40], which mainly contributed to lowering the water or fat release. In addition, da Silva et al. indicated that ionic interactions and the formation of hydrogen bonds between the side chain of L-Arg and water decreased the water/fat exudation of traditional bologna sausage [34].

### 3.4. Color Determination

The color of meat products is a critical factor that can intuitively influence consumer preference. As shown in Table 6, the PF group had significantly lower *L**- and *a**-values, as well as obviously higher *b**-value compared to those of the control group (*p* < 0.05). Choe et al. indicated that the decreased *L**-values and increased *b**-values of emulsion-type sausages without phosphates were mainly attributed to their lower pH values and higher cooking loss [4]. Meanwhile, the same amount of sodium nitrite was added to frankfurters of each group. However, the lower pH value (acidic condition) efficiently accelerated the decomposition of sodium nitrite [41], which probably altered the color parameters of frankfurters of the PF group. Moreover, the incorporation of 0.15% PC alone (T0 group) obviously increased the *L**-value and decreased the *a**- and *b**-values compared with those of phosphate-free frankfurters (PF group) (*p* < 0.05). When L-Arg was added in combination with PC, except for the sample with the addition of 0.025% L-Arg (T1 group), all other samples showed obviously higher *L**-values than those of the T0 group (*p* < 0.05). However, no significant differences were noted among the samples of the T2, T3, T4, and T5 groups (*p* > 0.05). Furthermore, on increasing the concentration of L-Arg, the *b**-values of phosphate-free frankfurters significantly decreased (*p* < 0.05). As mentioned above, the additive effects of L-Arg and PC obviously enhanced the pH values of phosphate-free frankfurters, which efficiently decreased the oxidation rate of myoglobin to metmyoglobin [42]. Ning et al. [43] indicated that L-Arg exhibited synergistic effects with sodium nitrite to promote the formation of nitrosylmyoglobin (a kind of typically cured pigment) and decrease the content of metmyoglobin, which subsequently led to lowering the *b**-values of sausage. In addition, the higher addition levels (0.1% and 0.125%) of L-Arg significantly increased the *a**-values of phosphate-free frankfurters when compared with those of the T0 group (*p* < 0.05). Given the excellent antioxidant capacities of L-Arg (particularly its metal chelation activity), the oxidation of oxy-myoglobin induced by metallic ions was effectively inhibited [35], which subsequently enhanced the *a**-values of phosphate-free frankfurters with higher levels of L-Arg. Ning et al. also indicated that the addition of L-Arg efficiently promoted the conversion of myoglobin to nitrosylmyoglobin and then enhanced the *a**-values of sausage [44].

The Δ*E**-value is an index used to evaluate the total color differences among different groups by accounting for the combined changes in *L**-, *a**-, and *b**-values [45,46]. A higher Δ*E**-value represents larger color changes compared to the control sample. As depicted in Table 6, the Δ*E**-value of the PF group was higher than 3, which indicated that the color difference between the control and PF groups have been perceived even by an inexperienced observer. Brainard also indicated that the color differences were prone to detected by an observer when the Δ*E**-value ranged from 2 to 10 [47]. Moreover, as the levels of L-Arg increased, the Δ*E**-value of phosphate-free frankfurters obviously decreased (*p* < 0.05). In particular, in the T4 and T5 groups, the Δ*E**-values were lower than 1, indicating that the color was difficult to be distinguished from that of the control group by an experienced observer. Therefore, our results indicated that the combination of L-Arg and PC could improve the color acceptance of phosphate-free frankfurters.

### 3.5. Texture

As shown in Table 7, because of the absence of phosphates, the PF group exhibited obviously higher hardness, as well as significantly lower resilience, springiness, chewiness, fracturability, and adhesiveness than the control group (*p* < 0.05). Our results were similar with the study of Schutte, Marais, Muller, and Hoffman, who reported that the removal of phosphates generated harder and more compact textures of meat products [48]. The reason for this quality deterioration was mainly attributed to the higher amount of water or fat release during the processing of emulsified meat products [49]. Choe et al. also indicated that phosphates effectively promoted the extraction of myofibrillar proteins and subsequently facilitate the formation of more stable textures of meat products after heating treatment [4]. Moreover, the incorporation of 0.15% PC alone (T0 group) remarkably decreased the hardness, as well as obviously increased the springiness and chewiness than those of phosphate-free frankfurters (PF group) (*p* < 0.05), which mainly attributed to the higher ionic strength and pH values. However, compared with the control group, the T0 group showed obviously lower hardness, resilience, fracturability, chewiness, and adhesiveness (*p* < 0.05), indicating that PC alone did not completely replace phosphates to maintain the quality profile of frankfurters. Furthermore, when L-Arg was added in combination with PC, as the addition levels of L-Arg increased, the resilience, springiness, fracturability, and chewiness of phosphate-free frankfurters significantly increased comparison with those of the T0 group (*p* < 0.05). However, the hardness of phosphate-free frankfurters with the addition of L-Arg at each level was lower than that of the T0 group (*p* < 0.05), which was in accordance with the decreased cooking loss. Zhou et al. [35] indicated that L-Arg significantly enhanced the springiness and chewiness of non-phosphate pork sausage. The combination of L-Arg and PC deviated the pH values of phosphate-free frankfurters from the isoelectric point and increased the solubility of meat proteins, which effectively promoted the formation of a better gel matrix after heating [37]. Additionally, L-Arg was found to exert a stronger inhibitory effect on protein aggregation and protein refolding [50], which significantly promoted the gel properties of protein gels. In particular, the phosphate-free frankfurters containing 0.15% PC and 0.1% L-Arg (T4 group) showed similar springiness, fracturability, and adhesiveness to those of the control group (*p* > 0.05), which indicated that the textural properties of phosphate-free frankfurters can be improved by the additive effects of L-Arg with PC.

### 3.6. Electronic Nose Analysis

Electronic nose analysis is an effective method for assessing the aroma of meat products. As depicted in Figure 2A, the W6S, W1S, W5S, W2S, and W1W sensors exhibited strong responses to VOCs of all frankfurters, indicating that the frankfurters contained higher levels of hydrides, methyl compounds, nitrogen oxides, alcohols, aldehydes, ketones, and sulfide compounds. Moreover, the PF group received the stronger signals from the W1C, W3C, W2S, W2W, W3S, and W1S sensors and weaker signals from the W5S, W1W and W6S sensors than the control group, which indicated that the incorporation of phosphates promoted the generation and volatilization of aromatic constituents, benzene, and methyl compounds and inhibited the generation and volatilization of aromatic, broad-alcohol, sulph-chlor, methane-aliph, and broad-methane compounds. Furthermore, the incorporation of 0.15% PC alone (T0 group) rendered weaker signals from the W5S, W6S, and W1S sensors in response to VOCs of phosphate-free frankfurters (PF group), suggesting that PC inhibited the formation and release of nitrogen oxides, hydrides, and methyl compounds. When L-Arg was combined with PC, the W6S, W1S, W2S, and W3S sensors exhibited a declining value in response to VOCs of the T0 group, implying that the combination of L-Arg with PC inhibited the formation and release of hydrides, methyl compounds, alcohols, aldehydes, ketones, and long-chain alkanes. However, compared with the T0 group, the signals from the W5S and W5C sensors enhanced when L-Arg was combined with PC, and the highest signals were observed in the T5 group. This suggested that the incorporation of PC and L-Arg promoted the generation and volatilization of nitrogen oxides, and short-chain alkane aromatic components of phosphate-free frankfurters.

As presented in Figure 2B, the frankfurters from different groups were obviously separated on the PCA chart, the control, PF, T0, T1, and T2 groups were situated on the positive PC1 axis, whereas all the other groups were situated on the negative axis. The control group related to the W1W and W6S sensors was kept away from the PF group, illustrating that the sulfide and hydride content in the control group were higher than those in the PF group. Moreover, the PF, T0, T1, T2, and T3 groups were close to each other and related to the W3S, W2S, and W1S sensors, indicating that the contents of long-chain alkanes, alcohols, aldehydes, ketones, and methyl were similar in these groups. On the negative PC1 axis, the signals of W5S and W5C sensors in the T5 group were stronger than those in the T3 and T4 groups, indicating that the T5 group had higher amounts of nitrogen oxides and short-chain alkane aromatic components.

### 3.7. E-Tongue Analysis

E-tongue analysis is a precise method used to evaluate the flavor profile of meat products. It can efficiently eliminate the subjectivity of sensory evaluation by panelists. As shown in Figure 3A, the PF group had stronger signals of sourness, bitterness, astringency, aftertaste-A (aftertaste of astringency), aftertaste-B (aftertaste of bitterness), and saltiness than the control group. The increase in sourness in the PF group may have been caused by the removal of phosphates, and this result was in accordance with that of the pH value of meat products. Moreover, compared with the PF group, the incorporation of PC alone (T0 group) decreased the sourness, astringency, and aftertaste-A. Furthermore, the combination of L-Arg and PC obviously decreased the sourness, bitterness, astringency, and aftertaste-A significantly, as well as increased the aftertaste-B, umami, richness, and saltiness. Additionally, compared with the control group, the samples prepared with the combination of L-Arg and PC showed obvious higher aftertaste-B, umami, richness, and saltiness, as well as lower sourness, which indicated that the additive effect of L-Arg and PC clearly enhanced the aftertaste-B, umami, richness, and saltiness of phosphate-free frankfurters.

The results of PCA of the e-tongue from all the groups are presented in Figure 3B. The control, PF, T0, and T1 groups were situated on the negative PC1 axis, whereas the other groups were situated on the positive axis. On the negative axis, the T0 and T1 groups were far from the control and PF groups based on the differences in their astringency, aftertaste-A, and bitterness. On the positive axis, the T4 and T5 groups were far from the T2 and T3 groups based on their richness, which indicated that the combination of L-Arg and PC at higher levels (0.1% or 0.125%) led to higher richness than that of the other groups. Our findings suggested that e-tongue analysis effectively distinguished the differences in flavor profiles of phosphate-free frankfurters produced with the combination of L-Arg and PC.

### 3.8. Lipid Oxidation

Lipid oxidation is a crucial factor that can induce quality deterioration in meat products during storage. As shown in Table 8, the TBARS values of all the frankfurters were extremely significantly affected by different formulations and storage times, as well as with interactions between the two factors. As the storage time increased, the TBARS values of frankfurters from each group were significantly increased (*p* < 0.05), suggesting the occurrence of lipid oxidation. Moreover, compared with the control group, the PF group exhibited obviously higher TBARS values during the entire storage period (*p* < 0.05), which was mainly due to the absence of phosphates. Bae, Cho, and Jeong indicated that effectively retarding lipid oxidation was one of the primary functions of phosphates in emulsified meat products, although phosphates were not classified as antioxidants [51]. Kılıç et al. also reported that phosphates could potentially delay lipid oxidation in cooked meat products by binding metal ions that act as catalysts for oxidation [6]. Furthermore, the incorporation of 0.15% PC alone (T0 group) evidently rendered lower TBARS values than the PF group during the same storage time (*p* < 0.05). However, the TBARS values were higher than those of the control group (*p* < 0.05), which is mainly due to the increase in pH values by the addition of PC. Kanner and Doll indicated that increased pH values probably inhibited the release of “free iron” from iron-carrying proteins [52]. This “free iron” can react with hydrogen peroxide to generate hydroxyl radicals and ultimately promote the degree of lipid oxidation [53]. However, our results implied that a lower concentration (0.15%) of PC alone did not completely replace phosphates to retard lipid oxidation in phosphate-free frankfurters during storage. Additionally, L-Arg in combination with PC significantly decreased the TBARS values of phosphate-free frankfurters than those of the T0 group (*p* < 0.05), regardless of the L-Arg levels or storage days. This phenomenon was due to the stronger antioxidant activities of L-Arg, which effectively inhibited lipid oxidation in phosphate-free frankfurters to some extent. Xu, Zheng, Zhu, Li, and Zhou indicated that the addition of L-Arg efficiently retarded lipid oxidation in emulsion-type sausages, mainly through its higher radical-scavenging and metal-chelating activities [54]. Zhang, Guo, Peng, and Jamali also indicated that the concentration of basic amino acids (e.g., L-Arg, L-Lys, and L-His) and the time of oxidation were crucial factors to determine the ability to inhibit lipid oxidation in meat products [55]. Overall, our findings suggested that the combination of L-Arg and PC could serve as an efficient functional ingredient to retard lipid oxidation in phosphate-free frankfurters.

### 3.9. Sensory Evaluation

The sensory attributes of frankfurters with or without phosphates were assessed in terms of the interior color, homogeneous, juiciness, and flavor. As shown in Table 9, the PF group exhibited lower sensory scores compared with the control group (*p* < 0.05), which indicated that the absence of phosphates resulted in severe quality defects in frankfurters. Petracci, Bianchi, Mudalal, and Cavani indicated that phosphates were considered vital functional additives for enhancing the water-holding capacity, cooking yield, and quality profile of meat products [56]. O’Flynna, Cruz-Romero, Troy, Mullen, and Kerry also reported that eliminating phosphates obviously inhibited the formation of the meat protein gel network, which led to a direct decrease in the scores of juiciness, flavor, firmness, and texture of breakfast sausages [57]. Moreover, compared with the PF group, the incorporation of 0.15% PC alone (T0 group) had remarkably higher scores of interior color, homogeneous, and flavor (*p* < 0.05). However, no significant differences were noted in the scores of juiciness between the PF and the T0 groups (*p* > 0.05). On the other hand, the T0 group also clearly exhibited lower sensory scores than the control group (*p* < 0.05). Our results indicated that PC alone did not completely replace phosphates without compromising the palatability of phosphate-free frankfurters. Furthermore, when L-Arg was added in combination with PC, except for the homogeneous scores, the scores of interior color, juiciness, and flavor were initially, gradually enhanced and then decreased with an increase in the addition levels of L-Arg (*p* < 0.05), and T4 group had the highest sensory scores among all the samples of phosphate-free frankfurters (*p* < 0.05). Meanwhile, the T4 group had similar score of uniformity with the control group (*p* > 0.05), which implied that combination of 0.1% L-Arg and 0.15% PC completely replaced 0.4% phosphates and successfully avoid some negative effects on the sensory properties of phosphate-free frankfurters to the maximum extent. Zhou et al. also reported that the incorporation of L-Arg efficiently masked the sensory defects in non-phosphate sausages [35]. In addition, the differences in sensory properties induced by different phosphate-replacing strategies are evident in the representative cross-sectional images of frankfurters from different groups. As shown in Figure 4, the combination of L-Arg and PC could overcome the quality defects and promote the sensory scores of phosphate-free frankfurters to some extent. However, the occurrence of some small voids in the structure of frankfurters seems to be a new quality defect which should be addressed in the future.

## 4. Conclusions

Successfully replacing phosphates without compromising the quality characteristics of emulsified meat products is challenging. Given its efficient pH-raising ability, PC has been considered a potential alternative to phosphates for overcoming quality defects in phosphate-free frankfurters. However, lower levels of PC could not exhibit the optimal phosphate-replacing effect, whereas higher levels of PC had adverse effects on the quality profile of the final products. The present finding indicated that compared with PC alone, the combination of L-Arg and PC could alleviate the quality defects and retard lipid oxidation in phosphate-free frankfurters more effectively and positively. In particular, 0.1% L-Arg combined with 0.15% PC was found to exhibit the best phosphate-replacing effect without increasing the manufacturing cost of phosphate-free frankfurters, except for the relatively softer textures (lower hardness, resilience, or chewiness) than those in the presence of phosphates, which was still a technological issue that should be addressed. Therefore, the phosphate replacement strategy proposed in the present study could be considered a viable alternative for the production of phosphate-free frankfurters with an improved quality profile and superior health benefits. Further research will focus on evaluating the formation of polycyclic aromatic hydrocarbons or biogenic amines and addressing the small voids in phosphate-free frankfurters when replacing phosphates with a L-Arg and PC.

## Figures and Tables

**Figure 1 foods-11-03581-f001:**
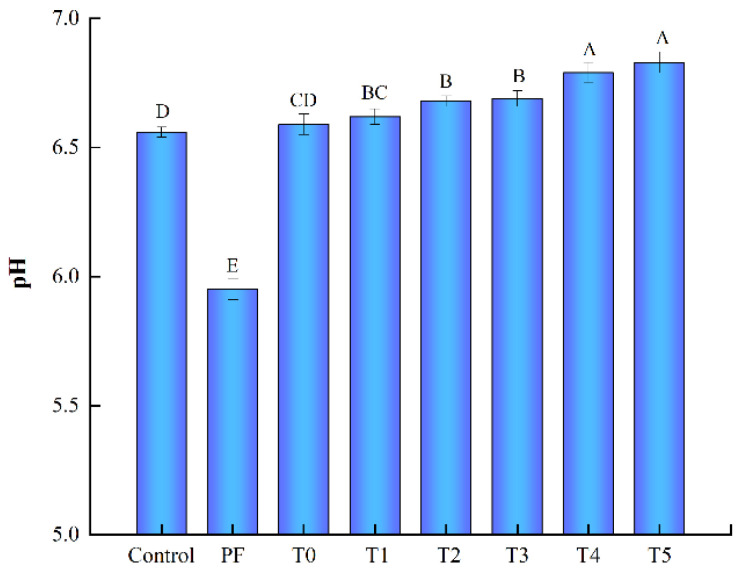
pH values of phosphate-containing frankfurters or phosphate-free frankfurters added with the combination of L-Arginine (L-Arg) and potassium carbonate (PC). The different uppercase letters (A–E) represent statistically significant differences for the different samples (*p* < 0.05). PF group: no added phosphate. T0 group: 0.15% PC and no added phosphate. T1 group: 0.15% PC + 0.025% L-Arg, and no added phosphate. T2 group: 0.15% PC + 0.05% L-Arg, and no added phosphate. T3 group: 0.15% PC + 0.075% L-Arg, and no added phosphate. T4 group: 0.15% PC + 0.1% L-Arg, and no added phosphate. T5 group: 0.15% PC + 0.125% L-Arg, and no added phosphate.

**Figure 2 foods-11-03581-f002:**
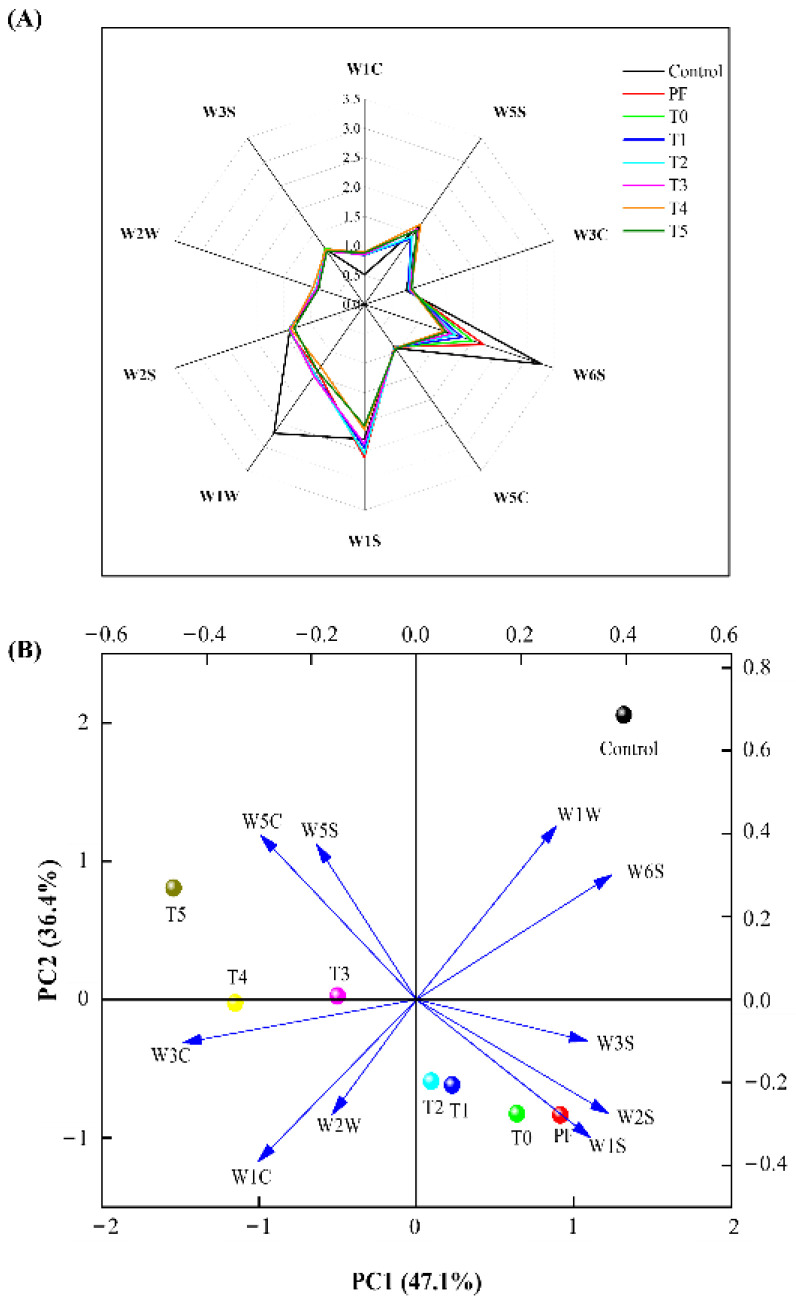
Radar chart (**A**) and principal component analysis (**B**) of the electronic nose data for phosphate-containing frankfurters or phosphate-free frankfurters added with the combination of L-Arg and PC. PF group: no added phosphate. T0 group: 0.15% PC and no added phosphate. T1 group: 0.15% PC + 0.025% L-Arg, and no added phosphate. T2 group: 0.15% PC + 0.05% L-Arg, and no added phosphate. T3 group: 0.15% PC + 0.075% L-Arg, and no added phosphate. T4 group: 0.15% PC + 0.1% L-Arg, and no added phosphate. T5 group: 0.15% PC + 0.125% L-Arg, and no added phosphate.

**Figure 3 foods-11-03581-f003:**
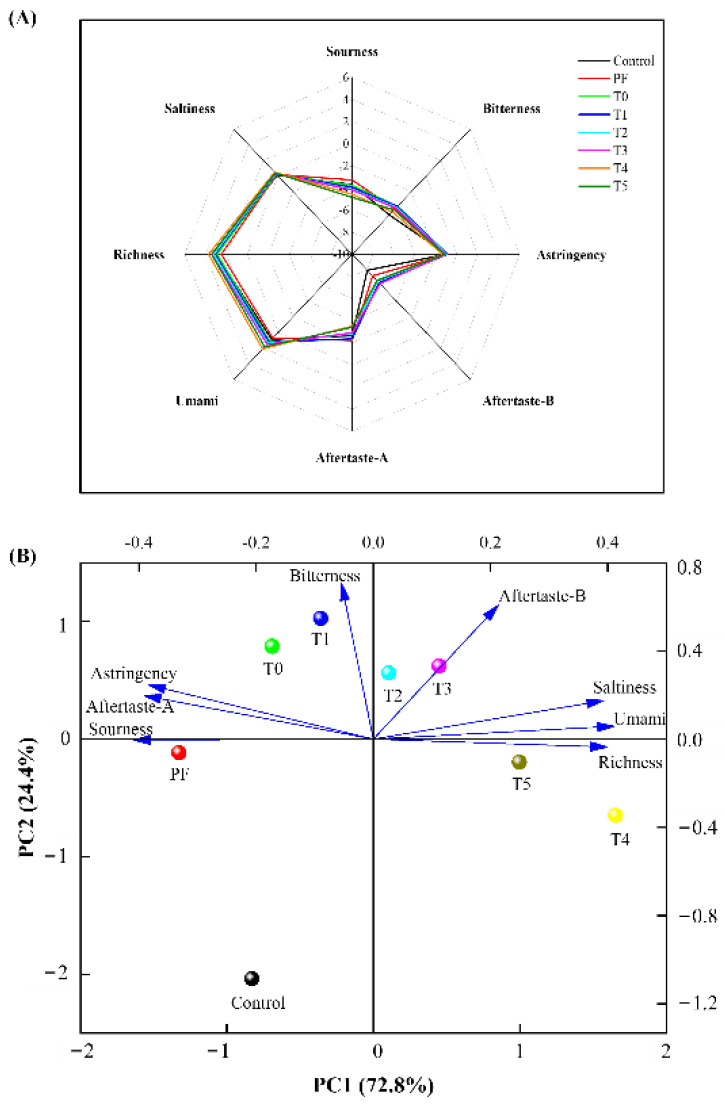
Radar chart (**A**) and principal component analysis (**B**) of the electronic tongue data for phosphate-containing frankfurters or phosphate-free frankfurters added with the combination of L-Arg and PC. PF group: no added phosphate. T0 group: 0.15% PC and no added phosphate. T1 group: 0.15% PC + 0.025% L-Arg, and no added phosphate. T2 group: 0.15% PC + 0.05% L-Arg, and no added phosphate. T3 group: 0.15% PC + 0.075% L-Arg, and no added phosphate. T4 group: 0.15% PC + 0.1% L-Arg, and no added phosphate. T5 group: 0.15% PC + 0.125% L-Arg, and no added phosphate.

**Figure 4 foods-11-03581-f004:**
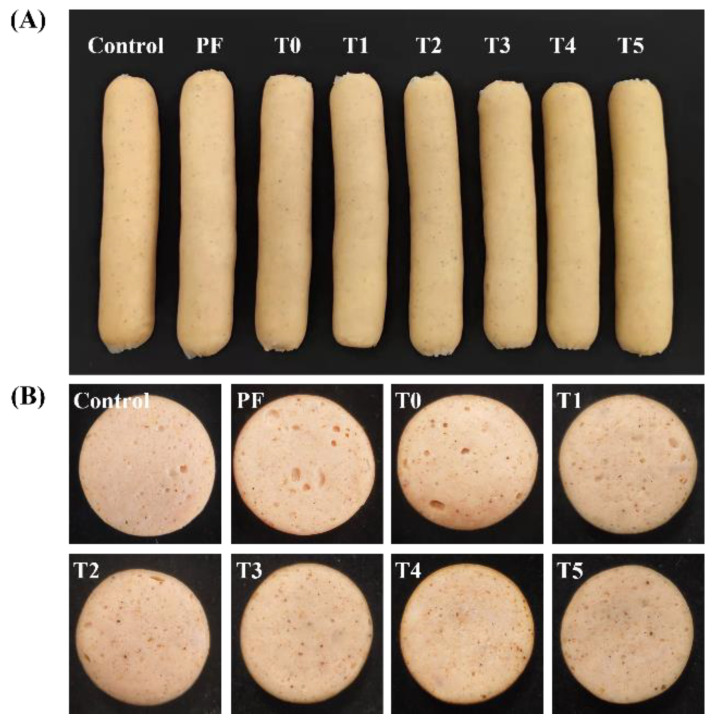
Images of phosphate-containing frankfurters or phosphate-free frankfurters added with the combination of L-Arg and PC. (**A**) the entire frankfurters. (**B**) cross section of frankfurters. PF group: no added phosphate. T0 group: 0.15% PC and no added phosphate. T1 group: 0.15% PC + 0.025% L-Arg, and no added phosphate. T2 group: 0.15% PC + 0.05% L-Arg, and no added phosphate. T3 group: 0.15% PC + 0.075% L-Arg, and no added phosphate. T4 group: 0.15% PC + 0.1% L-Arg, and no added phosphate. T5 group: 0.15% PC + 0.125% L-Arg, and no added phosphate.

**Table 1 foods-11-03581-t001:** Formulations (%) of phosphate-containing frankfurters or phosphate-free frankfurters added with the combination of L-Arginine (L-Arg) and potassium carbonate (PC).

	Control	PF	T0	T1	T2	T3	T4	T5
Ingredients								
Lean meat	50	50	50	50	50	50	50	50
Pork back-fat	25	25	25	25	25	25	25	25
Ice	25	25	25	25	25	25	25	25
Total	100	100	100	100	100	100	100	100
Additives *								
Sodium chloride	1.50	1.50	1.50	1.50	1.50	1.50	1.50	1.50
Sodium nitrite	0.0075	0.0075	0.0075	0.0075	0.0075	0.0075	0.0075	0.0075
Composite phosphates	0.4	0	0	0	0	0	0	0
Potassium carbonate	0	0	0.15	0.15	0.15	0.15	0.15	0.15
L-Arginine	0	0	0	0.025	0.050	0.075	0.100	0.125
Coriander seed powder	0.05	0.05	0.05	0.05	0.05	0.05	0.05	0.05
Macis powder	0.25	0.25	0.25	0.25	0.25	0.25	0.25	0.25
Red bell peppers powder	0.25	0.25	0.25	0.25	0.25	0.25	0.25	0.25
Ginger powder	0.30	0.30	0.30	0.30	0.30	0.30	0.30	0.30
White pepper powder	0.30	0.30	0.30	0.30	0.30	0.30	0.30	0.30
Monosodium glutamate	0.05	0.05	0.05	0.05	0.05	0.05	0.05	0.05
Sodium erythorbate	0.152	0.152	0.152	0.152	0.152	0.152	0.152	0.152

* The addition levels of additives were based on the total weight of lean pork meat, pork back fat, and ice. PF group: no added phosphate. T0 group: 0.15% PC and no added phosphate. T1 group: 0.15% PC + 0.025% L-Arg, and no added phosphate. T2 group: 0.15% PC + 0.05% L-Arg, and no added phosphate. T3 group: 0.15% PC + 0.075% L-Arg, and no added phosphate. T4 group: 0.15% PC + 0.1% L-Arg, and no added phosphate. T5 group: 0.15% PC + 0.125% L-Arg, and no added phosphate.

**Table 2 foods-11-03581-t002:** Information of 10 sensors for electronic nose.

Sensors	Representative Material Species	Description
W1C	Aromatic compounds	Sensitive to aromatic constituents, benzene
W5S	Broad range	Sensitive to nitrogen oxides
W3C	Aromatic	Sensitive to aroma, ammonia
W6S	Hydrogen	Sensitive to hydrides
W5C	Arom-aliph	Sensitive to short-chain alkane aromatic component
W1S	Broad-methane	Sensitive to methyl
W1W	Sulphur-organic	Sensitive to sulfides
W2S	Broad-alcohol	Sensitive to alcohols, aldehydes and ketones
W2W	Sulph-chlor	Sensitive to organic sulfides
W3S	Methane-aliph	Sensitive to long-chain alkanes

**Table 3 foods-11-03581-t003:** Attributes, definitions, scale anchors and standard references used in descriptive sensory analysis of frankfurters with different formulas.

Attribute	Definitions	Scale Anchors	Standard Reference
Interior color	Characteristic color of the frankfurters slice	1 = no pinkness, with light yellowness	Commercial frankfurters
		4 = light pinkness	
		7 = pinkness	
Homogeneous	Degree of uniformity of the frankfurters slice	1 = lower uniform and compact	Commercial frankfurters
		4 = intermediate uniform and compact	
		7 = higher uniform and compact	
Juiciness	Feeling of the moisture of the frankfurters during chewing	1 = extremely dry	Commercial frankfurters
		4 = juicy	
		7 = extremely juicy	
Flavor	Intensity of the characteristic flavor of the frankfurters, a mixture of meat and seasonings	1 = bland, not intense	Commercial frankfurters
		4 = neutral	
		7 = very intense	

**Table 4 foods-11-03581-t004:** Proximate compositions of phosphate-containing frankfurters or phosphate-free frankfurters added with the combination of L-Arg and PC.

	Moisture (%)	Protein (%)	Fat (%)	Ash (%)
Control	59.36 ± 0.13 ^A^	15.94 ± 0.14 ^A^	21.03 ± 0.29 ^C^	3.57 ± 0.12 ^A^
PF	58.08 ± 0.05 ^D^	14.15 ± 0.31 ^D^	19.46 ± 0.31 ^E^	3.15 ± 0.04 ^B^
T0	58.59 ± 0.05 ^C^	15.41 ± 0.28 ^C^	20.54 ± 0.15 ^D^	3.26 ± 0.13 ^B^
T1	58.69 ± 0.14 ^C^	15.58 ± 0.12 ^BC^	21.17 ± 0.04 ^C^	3.31 ± 0.22 ^B^
T2	58.87 ± 0.25 ^BC^	15.61 ± 0.03 ^ABC^	21.42 ± 0.25 ^BC^	3.29 ± 0.23 ^B^
T3	58.96 ± 0.18 ^B^	15.72 ± 0.12 ^ABC^	21.71 ± 0.41 ^AB^	3.25 ± 0.17 ^B^
T4	59.08 ± 0.07 ^B^	15.72 ± 0.21 ^ABC^	21.78 ± 0.46 ^AB^	3.19 ± 0.23 ^B^
T5	59.06 ± 0.21 ^B^	15.79 ± 0.11 ^AB^	21.91 ± 0.15 ^A^	3.14 ± 0.25 ^B^

Values are given as means ± SD from triplicate determinations; ^A–E^ in each column represent statistically significant differences (*p* < 0.05). PF group: no added phosphate. T0 group: 0.15% PC and no added phosphate. T1 group: 0.15% PC + 0.025% L-Arg, and no added phosphate. T2 group: 0.15% PC + 0.05% L-Arg, and no added phosphate. T3 group: 0.15% PC + 0.075% L-Arg, and no added phosphate. T4 group: 0.15% PC + 0.1% L-Arg, and no added phosphate. T5 group: 0.15% PC + 0.125% L-Arg, and no added phosphate.

**Table 5 foods-11-03581-t005:** Cooking loss and emulsion stability of phosphate-containing frankfurters or phosphate-free frankfurters added with the combination of L-Arg and PC.

	Cooking loss (%)	Emulsion Stability		
		Total Released Liquid (%)	Released Water (%)	Released Fat (%)
Control	7.47 ± 0.36 ^B^	6.50 ± 0.83 ^B^	6.01 ± 0.79 ^B^	0.48 ± 0.05 ^B^
PF	22.18 ± 0.24 ^A^	21.35 ± 0.76 ^A^	19.02 ± 0.53 ^A^	2.33 ± 0.23 ^A^
T0	6.93 ± 0.11 ^C^	6.45 ± 0.22 ^B^	5.95 ± 0.17 ^B^	0.50 ± 0.03 ^B^
T1	6.75 ± 0.09 ^CD^	6.38 ± 0.30 ^B^	5.89 ± 0.25 ^B^	0.49 ± 0.08 ^B^
T2	6.65 ± 0.20 ^CD^	5.89 ± 0.65 ^B^	5.46 ± 0.61 ^B^	0.44 ± 0.04 ^BC^
T3	6.39 ± 0.15 ^D^	5.61 ± 0.23 ^B^	5.20 ± 0.23 ^B^	0.41 ± 0.01 ^BC^
T4	5.51 ± 0.27 ^E^	4.24 ± 0.31 ^C^	3.92 ± 0.30 ^C^	0.32 ± 0.01 ^C^
T5	5.36 ± 0.05 ^E^	4.11 ± 0.58 ^C^	3.79 ± 0.55 ^C^	0.31 ± 0.03 ^C^

Values are given as means ± SD from triplicate determinations; ^A–E ^ in each column represent statistically significant differences (*p* < 0.05). PF group: no added phosphate. T0 group: 0.15% PC and no added phosphate. T1 group: 0.15% PC + 0.025% L-Arg, and no added phosphate. T2 group: 0.15% PC + 0.05% L-Arg, and no added phosphate. T3 group: 0.15% PC + 0.075% L-Arg, and no added phosphate. T4 group: 0.15% PC + 0.1% L-Arg, and no added phosphate. T5 group: 0.15% PC + 0.125% L-Arg, and no added phosphate.

**Table 6 foods-11-03581-t006:** Color of phosphate-containing frankfurters or phosphate-free frankfurters added with the combination of L-Arg and PC.

	*L* ***	*a* ***	*b* ***	Δ*E**
Control	63.56 ± 0.01 ^A^	13.45 ± 0.02 ^A^	20.14 ± 0.02 ^E^	-
PF	62.13 ± 0.08 ^D^	13.19 ± 0.07 ^B^	24.21 ± 0.07 ^A^	4.34 ± 0.18 ^A^
T0	62.40 ± 0.05 ^C^	12.52 ± 0.05 ^D^	23.55 ± 0.11 ^B^	3.50 ± 0.22 ^B^
T1	62.45 ± 0.02 ^C^	12.58 ± 0.20 ^D^	23.16 ± 0.11 ^C^	3.12 ± 0.07 ^C^
T2	62.56 ± 0.01 ^B^	12.77 ± 0.17 ^CD^	20.50 ± 0.06 ^D^	1.18 ± 0.13 ^D^
T3	62.58 ± 0.04 ^B^	12.79 ± 0.11 ^CD^	20.24 ± 0.03 ^E^	1.05 ± 0.14 ^DE^
T4	62.59 ± 0.07 ^B^	12.98 ± 0.24 ^BC^	19.92 ± 0.17 ^F^	0.92 ± 0.03 ^E^
T5	62.60 ± 0.02 ^B^	13.03 ± 0.08 ^B^	19.85 ± 0.07 ^F^	0.88 ± 0.01 ^E^

Values are given as means ± SD from triplicate determinations; ^A–F^ in each column represent statistically significant differences (*p* < 0.05). PF group: no added phosphate. T0 group: 0.15% PC and no added phosphate. T1 group: 0.15% PC + 0.025% L-Arg, and no added phosphate. T2 group: 0.15% PC + 0.05% L-Arg, and no added phosphate. T3 group: 0.15% PC + 0.075% L-Arg, and no added phosphate. T4 group: 0.15% PC + 0.1% L-Arg, and no added phosphate. T5 group: 0.15% PC + 0.125% L-Arg, and no added phosphate. *L**: lightness, *a**: redness/greenness, *b**: yellowness/blueness, Δ*E**: the total color difference.

**Table 7 foods-11-03581-t007:** Textural properties of phosphate-containing frankfurters or phosphate-free frankfurters added with the combination of L-Arg and PC.

	Hardness (g)	Resilience (g)	Springiness (%)	Fracturability (g)	Chewiness (g∙s)	Adhesiveness (g∙s)
Control	48.68 ± 0.73 ^B^	35.18 ± 0.72 ^A^	69.08 ± 0.57 ^ABC^	348.15 ± 12.64 ^A^	807.86 ± 15.23 ^A^	24.40 ± 1.15 ^A^
PF	51.55 ± 0.69 ^A^	27.16 ± 1.26 ^D^	64.27 ± 0.42 ^D^	283.03 ± 8.02 ^D^	642.61 ± 8.69 ^E^	19.25 ± 1.03 ^D^
T0	46.12 ± 2.00 ^C^	25.39 ± 1.38 ^D^	67.94 ± 0.70 ^C^	287.49 ± 3.25 ^D^	676.07 ± 14.68 ^D^	19.33 ± 0.37 ^D^
T1	43.60 ± 0.07 ^D^	26.68 ± 0.96 ^D^	68.12 ± 0.70 ^BC^	291.65 ± 7.58 ^D^	692.98 ± 18.71 ^CD^	19.72 ± 0.60 ^D^
T2	41.59 ± 0.82 ^DE^	29.39 ± 0.57 ^C^	69.55 ± 0.98 ^AB^	305.27 ± 6.80 ^C^	717.41 ± 17.79 ^C^	21.60 ± 0.84 ^C^
T3	41.36 ± 1.40 ^E^	29.71 ± 0.25 ^C^	69.75 ± 1.74 ^A^	328.27 ± 14.27 ^B^	722.97 ± 16.99 ^C^	23.93 ± 0.93 ^AB^
T4	40.70 ± 1.30 ^E^	30.35 ± 1.17 ^BC^	70.38 ± 0.81 ^A^	347.64 ± 8.55 ^A^	772.44 ± 23.42 ^B^	24.51 ± 0.83 ^A^
T5	39.88 ± 0.59 ^E^	32.22 ± 0.50 ^B^	70.61 ± 0.93 ^A^	335.62 ± 3.31 ^AB^	815.36 ± 15.67 ^A^	22.69 ± 0.83 ^BC^

Values are given as means ± SD from triplicate determinations; ^A–E^ in each column represent statistically significant differences (*p* < 0.05). PF group: no added phosphate. T0 group: 0.15% PC and no added phosphate. T1 group: 0.15% PC + 0.025% L-Arg, and no added phosphate. T2 group: 0.15% PC + 0.05% L-Arg, and no added phosphate. T3 group: 0.15% PC + 0.075% L-Arg, and no added phosphate. T4 group: 0.15% PC + 0.1% L-Arg, and no added phosphate. T5 group: 0.15% PC + 0.125% L-Arg, and no added phosphate.

**Table 8 foods-11-03581-t008:** The thiobarbituric acid reactive substances (TBARS) value (mg malonaldehyde/kg sample) of phosphate-containing frankfurters or phosphate-free frankfurters added with the combination of L-Arg and PC during 21 days of storage.

	Storage Time (Days)			
	1 d	7 d	14 d	21 d
Control	0.355 ± 0.012 ^Ec^	0.448 ± 0.010 ^Db^	0.463 ± 0.010 ^Fb^	0.493 ± 0.017 ^Fa^
PF	0.445 ± 0.013 ^Ad^	0.588 ± 0.010 ^Ac^	0.623 ± 0.005 ^Ab^	0.733 ± 0.005 ^Aa^
T0	0.425 ± 0.003 ^Bd^	0.517 ± 0.005 ^Bc^	0.597 ± 0.010 ^Bb^	0.623± 0.005 ^Ca^
T1	0.407 ± 0.005 ^Cc^	0.505 ± 0.013 ^BCb^	0.595 ± 0.010 ^Ba^	0.605 ± 0.004 ^CDa^
T2	0.379 ± 0.007 ^Dd^	0.500 ± 0.014 ^BCc^	0.547 ± 0.018 ^Db^	0.593 ± 0.015 ^DEa^
T3	0.386 ± 0.006 ^Dc^	0.497 ± 0.012 ^BCb^	0.503 ± 0.014 ^Eb^	0.577 ± 0.005 ^Ea^
T4	0.383 ± 0.005 ^Dd^	0.493 ± 0.017 ^Cc^	0.575 ± 0.006 ^Cb^	0.623 ± 0.005 ^Ca^
T5	0.387 ± 0.005 ^Dd^	0.458 ± 0.013 ^Dc^	0.590 ± 0.014 ^BCb^	0.660 ± 0.018 ^Ba^

Values are given as means ± SD from triplicate determinations; ^A–F^ in each column represent statistically significant differences (*p* < 0.05). ^a–d^ in each row represent statistically significant differences (*p* < 0.05). PF group: no added phosphate. T0 group: 0.15% PC and no added phosphate. T1 group: 0.15% PC + 0.025% L-Arg, and no added phosphate. T2 group: 0.15% PC + 0.05% L-Arg, and no added phosphate. T3 group: 0.15% PC + 0.075% L-Arg, and no added phosphate. T4 group: 0.15% PC + 0.1% L-Arg, and no added phosphate. T5 group: 0.15% PC + 0.125% L-Arg, and no added phosphate.

**Table 9 foods-11-03581-t009:** Sensory evaluation of phosphate-containing frankfurters or phosphate-free frankfurters added with the combination of L-Arg and PC.

	Interior Color	Homogeneous	Juiciness	Flavor
Control	6.17 ± 0.41 ^A^	5.38 ± 0.38 ^A^	6.00 ± 0.35 ^A^	6.17 ± 0.26 ^A^
PF	2.38 ± 0.48 ^F^	3.08 ± 0.58 ^E^	3.30 ± 0.27 ^E^	2.50 ± 0.50 ^E^
T0	3.25 ± 0.10 ^E^	3.77 ± 0.22 ^D^	3.42 ± 0.11 ^E^	3.20 ± 0.22 ^D^
T1	3.50 ± 0.50 ^E^	4.53 ± 0.25 ^C^	3.50 ± 0.50 ^E^	3.38 ± 0.15 ^CD^
T2	3.90 ± 0.55 ^D^	4.88 ± 0.38 ^B^	4.17 ± 0.26 ^D^	3.83 ± 0.41 ^C^
T3	4.42 ± 0.58 ^C^	5.00 ± 0.32 ^B^	4.58 ± 0.38 ^C^	4.74 ± 0.49 ^B^
T4	5.58 ± 0.49 ^B^	5.43 ± 0.38 ^A^	5.33 ± 0.26 ^B^	5.75 ± 0.42 ^A^
T5	4.50 ± 0.55 ^C^	5.32 ± 0.33 ^A^	4.42 ± 0.49 ^C^	4.67 ± 0.61 ^B^

Values are given as means ± SD from triplicate determinations; ^A–F^ in each column represent statistically significant differences (*p* < 0.05). PF group: no added phosphate. T0 group: 0.15% PC and no added phosphate. T1 group: 0.15% PC + 0.025% L-Arg, and no added phosphate. T2 group: 0.15% PC + 0.05% L-Arg, and no added phosphate. T3 group: 0.15% PC + 0.075% L-Arg, and no added phosphate. T4 group: 0.15% PC + 0.1% L-Arg, and no added phosphate. T5 group: 0.15% PC + 0.125% L-Arg, and no added phosphate.

## Data Availability

Not applicable.

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
