# Peer review of "Additive Effects of L-Arginine with Potassium Carbonate on the Quality Profile Promotion of Phosphate-Free Frankfurters"

_foods, 2022, doi:10.3390/foods11223581_

Round 1

Reviewer 1 Report

Generally, the present study is quite novel and the manuscript is well-written. However, there is some minor improvement that needs to be addressed by the author as suggested below: 

1. Line 65-68: 'Thus, how to successfully avoid a lower cooking yield and poorer quality of meat products following the reduction or removal of phosphates without increasing the manufacturing cost remains a challenge for the meat industry' - need to be re-phrased!.

2. Line 197-210: The reference number of the ethical approval letter approved by the Human Ethical committee for conducting Sensory analysis should be declared and written!

  i. The present manuscript is relevant and interesting because the L-Arginine combined with potassium carbonate (PC) to replace phosphate in the formulation of frankfurters has improved its quality. The topic is also original. 

ii. Besides, the manuscript is also well written. The text is also clear and easy to read. 

iii. The conclusion section is also consistent with the evidence and arguments presented. 

iv. The author also has addressed the main question which is that L-Arg combined with PC is able to decrease cooking loss and improve textural properties of phosphates-free frankfurters.   v. However, the authors need to show human ethical clearance for permission of performing the sensory evaluation. Besides, the authors are also advised to include the health implication of processed meat-based products formulated with phosphates.

Author Response

Response to Reviewer 1#

Generally, the present study is quite novel and the manuscript is well-written. However, there is some minor improvement that needs to be addressed by the author as suggested below:

Q1: Line 65-68: 'Thus, how to successfully avoid a lower cooking yield and poorer quality of meat products following the reduction or removal of phosphates without increasing the manufacturing cost remains a challenge for the meat industry' - need to be re-phrased!.

A1: This is a good suggestion. According to the reviewer’s opinion, we have re-written this statement in our revised manuscript as follows:

“Thus, how to promote the quality profiles of reduced phosphates or phosphate-free processed meat products without increasing the manufacturing cost is challenging for meat industry.”

We are indebted to the reviewer for this constructive suggestion to improve the quality and readability of the manuscript. Thanks.

Q2: Line 197-210: The reference number of the ethical approval letter approved by the Human Ethical committee for conducting Sensory analysis should be declared and written!

A2: This is a good suggestion. First of all, we were apologized for our unclear statement about the detailed approval permission of the Ethics in Research Committee of Northeast Agricultural University. According to the reviewer’s opinion, we have added some statements as follows:

“The sensory evaluation was carried out under the approval permission of the Ethics in Research Committee of Northeast Agricultural University (ERCNEAU-2018-001). All the panelists were agreed voluntarily to participate in the sensory evaluation.”

We are indebted to the reviewer for this constructive suggestion to improve the quality of the manuscript. Thanks.

Q3: i. The present manuscript is relevant and interesting because the L-Arginine combined with potassium carbonate (PC) to replace phosphate in the formulation of frankfurters has improved its quality. The topic is also original. ii. Besides, the manuscript is also well written. The text is also clear and easy to read. iii. The conclusion section is also consistent with the evidence and arguments presented. iv. The author also has addressed the main question which is that L-Arg combined with PC is able to decrease cooking loss and improve textural properties of phosphates-free frankfurters. v. However, the authors need to show human ethical clearance for permission of performing the sensory evaluation. Besides, the authors are also advised to include the health implication of processed meat-based products formulated with phosphates.

A3: This is a good suggestion. First of all, we were apologized for our unclear statement about the detailed approval permission of the Ethics in Research Committee of Northeast Agricultural University. According to the reviewer’s opinion, we have added some statements as follows:

“The sensory evaluation was carried out under the approval permission of the Ethics in Research Committee of Northeast Agricultural University (ERCNEAU-2018-001). All the panelists were agreed voluntarily to participate in the sensory evaluation.”

Moreover, according to the reviewer’s opinion, we have added some statement about the health implication of processed meat-based products formulated with phosphates as follows:

“Due to the excessive intake amounts of phosphates from processed meat products contributes to an increased risk of health implication.”

We are indebted to the reviewer for this constructive suggestion to improve the quality of the manuscript. Thanks.

Reviewer 2 Report

Presented manuscript is very well-written and quality of paper is high. I recommend very few minor changes in it’s text.

Introduction

Line 29: change addi-tives to additives

Materials and Methods

In this chapter packaging and storage condition of meat products should be mentioned. Please provide the information.

2.1

Line 110 – 112: It would be more suitable to list fat and protein content of final meat product than meat itself.

2.2

Line 129 – 131: Was the content of fat and protein and other components alike in all 3 replications?

2.11

In sensory analysis, before the evaluation, were the samples heated, room temperature or from refrigerator?

Author Response

(The authors gave the same response as above.)
